# Dendritic Cells of Leukemic Origin (DC_leu_) Modulate the Expression of Inhibitory Checkpoint Molecules and Their Ligands on T Cells and Blasts in AML Relapse After Allogeneic Stem Cell Transplantation

**DOI:** 10.3390/cancers17182948

**Published:** 2025-09-09

**Authors:** Xiaojia Feng, Giuliano Filippini Velázquez, Sophia Bohlscheid, Marianne Unterfrauner, Philipp Anand, Hazal Aslan Rejeski, Anne Hartz, Tobias Baudrexler, Christoph Schmid, Helga Maria Schmetzer

**Affiliations:** 1Department of Medicine III, University Hospital of Munich, 81377 Munich, Germany; sophia.bohlscheid@gmx.de (S.B.);; 2Bavarian Center for Cancer Research (BZKF), 91054 Erlangen, Germany; giuliano.filippinivelazquez@uk-augsburg.de (G.F.V.); christoph.schmid@uk-augsburg.de (C.S.); 3Department of Hematology and Oncology, Augsburg University Hospital and Medical Faculty, 86156 Augsburg, Germany

**Keywords:** acute myeloid leukemia, allogeneic hematopoietic stem cell transplantation, immune checkpoint molecules, dendritic cell-based therapy

## Abstract

A total of 30–40% of AML patients relapse following allogeneic hematopoietic stem cell transplantation (allo-HCT) due to immune escape mechanisms. Granulocyte–Macrophage Colony-Stimulating Factor (GM-CSF) and Prostaglandin-E1 (PGE-1) (“Kit M”) convert blasts in patients’ whole blood (WB) into leukemia-derived dendritic cells (DC_leu_), enabling leukemia-specific immune reactivation after mixed lymphocyte culture (MLC). We quantified immune checkpoint molecule/ligand (ICM/ICML)-expressing immune effector cells and blasts before/after Kit M treatment, examining both the WB of relapsed AML patients after allo-SCT, and healthy WB—Kit-M-pretreated (vs. not-pretreated) leukemic blood improved blast lysis after MLC. Higher frequencies of ICM co-expressing uncultured T-cells/blasts correlated negatively with blast lysis after MLC or patients’ clinical response to relapse therapy. However, post-MLC, Kit-induced blast lysis was independent of the frequencies of ICM-expressing T cells. The quantification of ICM/ICML-expressing cells could contribute to an accurate assessment of the efficacy of AML immunotherapy in future clinical applications. Kit-M has the potential to target the exhaustion status of ICM/ICML-mediated immune cells, thereby enabling the reactivation of antileukemic immune cells. This DC/DC_leu_-mediated mechanism could contribute to improvements in therapy as well as the outcome in relapsed AML patients after allo-HCT.

## 1. Introduction

### 1.1. Acute Myeloid Leukemia Relapse After Allogeneic Hematopoietic Stem Cell Transplantation

Acute myeloid leukemia (AML) is a life-threatening hematological malignancy characterized by the clonal expansion of undifferentiated myeloid precursors [1]. Allogeneic hematopoietic stem cell transplantation (allo-HCT) remains the only curative treatment for patients with high-risk AML. However, relapses occur in 30–40% of AML patients after initial allo-HCT; these are associated with poor outcomes, with an estimated median survival of <6 months [2,3,4]. A better understanding of the biological mechanisms underlying leukemia relapse after allo-HCT is essential for optimizing treatment strategies [5].

### 1.2. Immune Checkpoint Molecules and Immune Escape

Immune escape in AML relapse after allo-HCT involves various mechanisms by which leukemia cells evade detection and destruction by the immune system [6,7,8,9]. In healthy organisms, immune checkpoint molecules (ICMs) are expressed on B, T, and NK cells and also on myeloid progenitor cells, monocytes, DCs, and mesenchymal stem cells (MSCs); these regulate immuno-responses as “natural brakes” to establish self-tolerance [10,11,12,13]. Sustained expression of inhibitory receptors represents an important immune evasion strategy exploited by cancer cells, making some of these molecules potential targets for immunotherapy [7]. In AML, upregulation of ICM on T/NK cells or their corresponding ligands (ICMLs) on blasts has been reported in up to 40% of relapses after allo-HCT, mediating immune escape [4]. Key ICMs implicated in AML immune escape include CTLA4, PD1, TIM3, TIGIT, KLRG1, LAG3, and 2B4.

Ourselves and other researchers have extensively reviewed the role of ICM/ICML-expressing cells and other mechanisms in AML relapse after allo-HCT [7,13].

### 1.3. DC-Based Immunotherapy

Dendritic cells (DCs) are specialized antigen-presenting cells (APCs) crucial for initiating immune responses. Upon activation by danger signals, such as nucleic acids or infectious particles, DCs upregulate molecules like chemokine receptors and major histocompatibility complex (MHC) antigens, enhancing their ability to activate the immune system [14,15].

DC-based strategies, including ex vivo generated monocyte-derived DCs loaded with leukemic antigens or fused leukemic blasts with DCs as vaccines, as well as leukemia-derived DCs (DC_leu_) infused to patients, have shown promising results in both clinical and immunological settings, making them attractive therapeutic options for AML patients [14,16].

In ex vivo experiments using T-cell-enriched mixed-lymphocyte culture (MLC) from AML patients’ WB, Kit M (containing Granulocyte–Macrophage Colony-Stimulating-Factor (GM-CSF) and Prostaglandin E1(PGE1))-induced DC_leu_, which present the patients’ specific leukemic antigens, has been shown to induce leukemia-specific/antileukemic immune responses [17,18,19]. In vivo experiments involving leukemic rats, as well as an experimental clinical treatment of highly refractory AML patients before or after allo-HCT, showed leukemia-specific immune cell activation and stabilization of the disease [20,21].

### 1.4. Innate and Adaptive Immune System and Antileukemic Activity

The immune system comprises both innate and adaptive components, each playing distinct roles in defending the body against pathogens and tumor cells. The innate immune system acts as the first line of defense and includes macrophages, monocytes, DCs, cytokine-induced killer cells (CIKs), and natural killer (NK) cells [17,18,22,23]. These cells respond rapidly to pathogens and tumors and are crucial for initiating immune responses. On the other hand, the adaptive immune system (B and T-cell subtypes) provides specific and long-term immunity [18,19,23]. Potentially leukemia-specific cells can be identified using the Intracellular Cytokine Assay (INCYT), quantifying intracellular IFNγ and TNF-α-producing immune cells. These immunoreactive cells and their subtypes can be detected using flow cytometry (Appendix A). Additionally, the Degranulation Assay (DEG) quantifies cells expressing lysosomally associated membrane glycoproteins (LAMPs) involved in perforin-associated degranulation processes. This assay provides detection and quantification of leukemia-specific cells and their activity. The Cytotoxicity Fluorolysis Assay (CTX) assesses the antileukemic effect of effector cells on viable blasts. By combining these assays, a comprehensive analysis of immune cells’ function and antileukemic activity can be obtained [17,18,19,23].

### 1.5. Aim of the Study

We hypothesized that modulation of immune escape mechanisms, particularly downregulation of ICM-expressing immune effector cells or blasts, could enhance antileukemic immunity, a concept that has not yet been extensively studied, especially in the context of DC_leu_ strategies. Using flow cytometry, we studied ICM/ICML expressing (uncultured) T-cells and blasts in peripheral blood from AML patients who relapsed after allo-HCT. We generated ex vivo DC_leu_ using Kit M and quantified ICM expressing DC/DC_leu_ subsets and ICM/ICML-expressing T-cells after MLC was stimulated with vs. without Kit M-pretreated patients’ WB. Finally, we correlated ICM/ICML-expressing uncultured T-cells/blasts and T-cells/blasts after MLC (of Kit M-pretreated vs. not pretreated WB) with the provision of leukemia-specific cells, ex vivo blasts’ lysis, patients’ allocation to risk groups, and clinical response to relapse treatment.

With this study, we contribute insights into the dynamics of AML relapse mechanisms and immune-mediated treatment resistance, particularly in the context of DC_leu_-based therapies for post-transplant relapse. With our work, we contribute to developing new treatment strategies that aim to overcome immune evasions in patients who relapse after SCT.

## 2. Materials and Methods

### 2.1. Sample Collection

Heparinized patients’ peripheral WB samples were collected prospectively at the University Hospital Augsburg at the time of hematological relapse after allo-HCT, before the initiation of relapse treatment. Sample collection was conducted after obtaining informed consent and in accordance with the Helsinki protocol. This study was approved by the ethics committee of the Ludwig-Maximilians University Hospital Munich (Vote-No 339-05).

### 2.2. Patients

Patients who had undergone allo-HCT for AML or MDS from a matched-sibling, matched-unrelated, mismatched unrelated, or haploidentical donor (MSD, MUD, Haplo), and subsequently developed hematological or molecular relapse were included in the study.

Conditioning regimens and Graft-vs.-Host Disease prophylaxis were performed according to institutional standards. Relapse was confirmed through comprehensive bone marrow analysis (BM), including cytomorphology, flow cytometry, donor chimerism assessment, cytogenetics, and molecular genetics, alongside the detection of blasts in peripheral blood (PB).

Per institutional standards, all patients with post-allo-HCT relapse were offered salvage treatment with approved agents, and a subsequent further allo-HCT or donor lymphocyte infusions (DLIs), based on response to salvage treatment, clinical performance status, and donor availability, taking patients’ personal preferences into consideration. Response to salvage treatment was assessed through repeated BM biopsies following international criteria (European Leukemia ELN 2022). Treatment response was categorized as complete remission (CR) or partial remission (PR), whereas no response was defined by clinical progression/refractoriness.

We examined whole blood (WB) samples from 15 patients with AML/MDS that were obtained at the time of hematological relapse after allo-HCT, and WB from 10 healthy volunteers. The median age of AML patients was 59 (range: 23–73) years, and for healthy volunteers, it was 37 (range: 22–59) years. Eleven patients relapsed after the first allo-HCT, and four relapsed after the second allo-HCT. The median time from allo-HCT to relapse was 13 months (range: 40–59). Salvage therapy consisted mainly of hypomethylating agents (decitabine or azacitidine) alone (*n* = 2) or in combination with venetoclax (*n* = 12). One patient received intensive chemotherapy consisting of cytarabine, mitoxantrone, and venetoclax. A response to salvage therapy was observed in 7/15 patients. More details can be found in Appendix A.

### 2.3. Flow Cytometry and Sample Preparation

To evaluate and quantify frequencies of ICM/ICML-expressing T-cells, Blasts, or DC, phenotypes of DC/DC_leu_, leukemic blasts, monocytes, and immune-reactive cell subsets of adaptive and innate immunity, flow cytometric analyses were performed using a fluorescence-activating cell-sorting flow cytometer FACS Calibur (BD, San Jose, CA, USA) [17,23]. WB samples from AML patients or healthy volunteers were used to set up DC cultures and to perform the Degranulation (Deg) Assay and the Intracellular Cytokine (InCyt) Assay directly after collection. Details of the technical procedures are provided in the Appendix A. The cell subtypes detected are listed in Appendix A.

### 2.4. Dendritic Cell Culture (DCC)

The generation of DC/DC_leu_ from healthy and leukemic WB was performed using specific response modifiers according to established DC/DC_leu_-generating protocols (Kit M) as shown before [17,18,19]. (Frequencies of ICM/ICML-expressing) DC/DC_leu_ subtypes were evaluated by flow cytometry. In cases involving less than 4.5% blasts in the cell fractions, DC_leu_ and associated subgroups could not be evaluated. Details of the technical procedures are provided in the Appendix A. The cell subtypes detected are listed in Appendix A.

### 2.5. T-Cell-Enriched Mixed Lymphocyte Culture (MLC) and Functional Analyses (Intracellular Cytokine Assay (InCyt), Degranulation Assay (DEG), Cytotoxicity Fluorolysis Assay (CTX))

To generate T cell-enriched immune-reactive cells, thawed patient or healthy T-cells were stimulated with Kit M-pretreated (or not pretreated) DC/DC_leu_ containing WB, as described previously [17,18,19]. Flow cytometric quantifications of ICM/ICML-expressing and intracellularly IFNγ-producing or -degranulating T-cell subsets were performed via flow cytometry, as described previously [17,18,24], were quantified using DEG and the InCyts (to quantify potentially leukemia-specific cells. These cells were quantified in uncultured patients’ blood samples after stimulation with leukemia-associated antigens (LAAs, WT1, and PRAME in AML patients’ samples and with Staphylococcal Enterotoxin B (SEB) in healthy blood samples). Leukemia-specific cells were also quantified after 7 days of stimulation of cells in T cell-enriched MLC with Kit-treated WB [17,18]. The cell subtypes detected are listed in Appendix A. The Cytotoxicity Fluorolysis Assay (CTX) was performed to analyze the blast lytic activity of T cell-enriched immune-reactive cells in MLC^WB-DC(Kit M)^ and MLC^WB-DC(Control)^, as described before [17]. Details of the technical procedures are provided in the Appendix A.

### 2.6. Statistical Methods

Statistical analyses and figures were performed using Excel 2022 (Microsoft, Redmond, WA, USA) and Prism 9 (GraphPad Software, San Diego, CA, USA). The data are presented as the mean ± standard deviation.

Data normality was assessed using the Shapiro–Wilk test. A *p*-value > 0.05 indicated that the data followed a normal distribution and justified the use of *t*-tests: Differences were considered as “highly significant” in cases with *p*-values ≤ 0.005, as “significant” with *p*-values ≤ 0.05, and as “borderline significant” with *p*-values between 0.05 and 0.10. In cases with *p* < 0.05, non-parametric alternatives (e.g., Wilcoxon signed-rank, with *p* < 0.05 considered statistically significant) were applied. In selected analyses, comparisons/correlations, the relative changes in frequencies of cell subsets (deltas) between Kit M-treated and untreated settings are provided.

## 3. Results

We quantified ICM/ICML-expressing uncultured immune cells from WB of healthy donors and immune cells/blasts from WB of AML/MDS patients that relapsed after allo-HCT (details are given in the Methods and in Appendix A), and ICM/ICML-expressing (immune) cells after DC/MLC. We correlated results with (ex vivo) functional analyses and patients’ clinical data.

### 3.1. ICM/ICML Expressing Uncultured Blasts and T-Cells from AML Patients at Relapse After Allo-HCT, and on Immune Cells from Healthy Donors

#### 3.1.1. High Frequencies of Uncultured AML-Blasts Co-Express ICM (CTLA4 and PD1)

Eight out of fifteen AML/MDS patients presented with an average (Ø) of 15.94 (range: 4.5–61) % PB-blasts (seven cases with ≤4% blasts in PB were excluded from this analysis). In the blast fraction, we found high frequencies of PD1 (Ø 22.03 ± 30.76% BLA_PD1+_/BLA+) and CTLA4 (Ø 20.67 ± 28.28% BLA_CTLA4+_/BLA+) co-expressing blasts. Two patients presented with >60% BLA_CTLA4+_/BLA+ and BLA_PD1+_/BLA+, respectively, leading to Ø 1.5% of PD1 or CTLA+ blasts in the WB fraction (BLA_CTLA4+_/WB and BLA_PD1+_/WB). Low frequencies of blasts co-expressing the ICML PDL1 and PDL2 were found. More data are given in Figure 1(A1,A2) and Appendix A.

#### 3.1.2. Higher Frequencies of Uncultured T-Cell (Subtypes) from AML Patients vs. Healthy Donors Co-Express ICM (CTLA4 and PD1)

Patients’ WB samples (*n* = 15) presented with Ø 11 (range: 3–19)% T-cells. We found high frequencies of T_CD3+_ (and especially T_CD3+CD4−_) cells co-expressing CTLA4 (Ø 51.94 ± 30.48% T_CD3+CTLA4+_/T_CD3+_) and high frequencies of T_CD3+_ cells (especially T_CD3+CD4+_ cells) co-expressing PD1 (Figure 1B, upper part, left side). This was not the case for healthy donors’ T-cells, except for one healthy donor, who presented with 97.12% T_CD3+CTLA4+_/T_CD3+_ and 97.4% T_CD3+PD1+_/T_CD3+_ cells (Figure 1B, upper part, right side).

We found significantly more CTLA4-expressing T_CD3+_ cells in AML compared to healthy donors’ T-cells, with 13 of the 15 patients’ samples presenting with >20% of T_CD3+CTLA4+_/T_CD3+_. Furthermore, 8 out of 10 patients demonstrated (significantly) elevated frequencies of T_CD3+CD4−_ cells co-expressing TIGIT (T_CD3+CD4−TIGIT+_/T_CD3+CD4−_) compared to healthy samples (Figure 1B, lower part).

### 3.2. Increased Frequencies of (Antigen-Specific) Intracellular IFNγ-Producing or Degranulating Immune Cells After Stimulation of Uncultured WB (AML and Healthy) with vs. Without LAA/SEB

Frequencies of adaptive and innate/antigen-/leukemia-specific immune cells (e.g., degT_non-naïve_/T_non-naïve_, degT_cm/_T_cm_, degCIK/CIK) were increased (or decreased: degT_CD4+reg_/T_CD4+reg_) after the addition of leukemia-specific (WT1, PRAME) to AML or Staphylococcal Enterotoxin B (SEB) antigens to healthy donors’ samples, as shown previously [18,25] (Appendix A).

### 3.3. Characteristics of DC from AML and Healthy WB After Culture with (vs. Without) Kit M

#### 3.3.1. Increased Frequencies of (Leukemia)-Derived DC/DC_leu_ in AML or Healthy WB After Culture with (vs. Without) Kit M

Frequencies of (mature/leukemia-derived) DC (e.g, DC_mat_/WB, DC_mat+leu_/WB) were significantly increased under the influence of Kit M (vs. control) without induction of blast (in AML) or monocyte (in healthy patients) proliferation, thereby confirming data shown before [18,19] (Appendix A).

#### 3.3.2. Decreased Frequencies of ICM/ICML Expressing DC from AML WB After Culture with (vs. Without) Kit M

Following Kit M (vs. without) treatment of WB, frequencies of ICM/ICML (esp. CTLA4, 2B4) expressing DC were (borderline significantly) downregulated (e.g., %DC_CTLA4+_/DC: Kit M: 24.06 ± 12.33 vs. control 34.82 ± 18.17, *p* = 0.06, Figure 2). Analogously, we quantified ICM/ICML expressing DC_leu_ subtypes. Evaluation of ICM/ICML-expressing DC_leu_ after Kit M treatment was infeasible due to low blast/DC_leu_ counts.

### 3.4. Characteristics of T-Cells After MLC with Kit M-Pretreated (vs. Not Pretreated) Patients’ and Healthy Donors’ WB

#### 3.4.1. Activated and Memory T-Cells Increased After MLC (With Kit M-Pretreated vs. Not Pretreated Patients’ and Healthy Donors’ WB)

Frequencies of activated/memory (T_em_/T_cm_) cells were significantly upregulated after T cell-enriched MLC with Kit M-pretreated vs. not pretreated patients’ or healthy donors’ WB (Appendix A).

#### 3.4.2. Decreased Frequencies of ICM/ICML-Expressing T-Cells After MLC of Kit M-Pretreated vs. Not Pretreated AML Patients’ and Healthy Donors’ Samples

Frequencies of ICM-expressing T_CD3+_ cells (e.g., T_CD3+CTLA4+/_T_CD3+_, T_CD3+PD1+/_T_CD3+_, T_CD3+TIGIT+/_T_CD3+_) were significantly downregulated after MLC with Kit M-pretreated vs. not pretreated AML patients’ and healthy donors’ samples (Figure 3).

### 3.5. Increased Frequencies of (Antigen-Specific) Intracellular IFNγ-Producing or -Degranulating Immune-Activated and Memory T or Innate Cells After MLC with Kit M-Pretreated (vs. Not Pretreated) Patients’ or Healthy Donors’ WB

Frequencies of intracellular IFNγ-producing and degranulating activated/memory T-cells (e.g., IFNγ + T_em/eff_/T_em/eff_ or IFNγ + T_cm_/T_cm_; degT_em/eff_/T_em/eff_ or degT_cm_/T_cm_) or innate cells (e.g., IFNγ + NK/NK, degCIK/CIK) were significantly upregulated after T-cell-enriched MLC with Kit M-pretreated vs. not pretreated (healthy or leukemic) WB, as already shown [17,19] (Appendix A).

### 3.6. Increased Antileukemic Cytotoxicity After MLC in Kit-M-Treated AML Patients’ WB

With our analyses, we confirm findings that have already been demonstrated before [17,19]: blast kill is improved using samples pretreated (vs. not pretreated) with Kit M as effector cells (Appendix A).

### 3.7. Correlations of Frequencies of ICM/ICML-Expressing Cells with Ex Vivo Achieved Blast Lysis or Patients’ Response to Relapse Treatment

We correlated frequencies of uncultured ICM/ICML-expressing T-cells or blasts; of ICM/ICML-expressing DC (after DC culture); and of ICM/ICML-expressing T-cells after MLC with ex vivo achieved blast lysis, as well as with patients’ clinical/hematological characteristics.

#### 3.7.1. Low Frequencies of ICM/ICML-Expressing Uncultured AML Patients’ T-Cells Correlate with Ex Vivo Achieved Blast Lysis After MLC, and with Clinical Response to Initial Salvage Therapy

In correlative univariable analyses (*t*-test), in uncultured patients’ WB, we observed significant correlations between low frequencies of uncultured T-cells co-expressing ICM and ex vivo achieved blast lysis after MLC (e.g.: T_CD3+CTLA4+_/T_CD3+_ (*p* = 0.03), T_CD3+PD1+_/T_CD3+_ (*p* = 0.05), T_CD3+2B4+_/T_CD3+_ (*p* = 0.02), T_CD3+TIM3+_/T_CD3+_ (*p* = 0.08) (Figure 4A and Appendix A). Furthermore, patients with lower frequencies of T_CD3+CTLA4+_/T_CD3+_ and T_CD3+TIGIT+_/T_CD3+_ showed a better clinical response to salvage treatment (Figure 4C).

#### 3.7.2. No Correlations of Frequencies of ICM/ICML-Expressing T-Cells After MLC of Kit M-Pretreated vs. Not Pretreated AML Patients with Ex Vivo Achieved Blast Lysis or with Patients’ Response to Relapse Treatment

We performed correlative univariable analyses (*t*-test) of ICM-expressing T-cells and the rate of ex vivo achieved blast lysis after MLC using Kit M-pretreated vs. untreated patients’ WB. We did not observe any correlations between frequencies of T-cells co-expressing ICM and ex vivo achieved blast lysis (Figure 4B and Appendix A) despite reduced frequencies of several ICM-expressing T-cells after MLC. There were also no correlations with the clinical response to salvage treatment in patients.

In summary, we

(1) Observed high frequencies of various ICM-expressing (uncultured) blasts and patients’ T-cells.

(2) Observed decreased frequencies of ICM-expressing T-cells in AML/MDS patients’ blood samples after treatment with Kit M and MLC.

(3) Observed ex vivo improved blast lysis in cases with lower frequencies of (especially CTLA4, PD1, 2B4, and TIM3) ICM-expressing uncultured T-cells.

(4) Observed higher rates of patients with successful salvage therapy presenting with ex vivo lower frequencies of CTLA4-expressing uncultured T-cells in patients.

(5) Did not observe improved blast lysis or patients’ response to salvage therapy despite lower frequencies of ICM/ICML-expressing T-cells after MLC.

(6) Confirmed previous findings that Kit M pretreatment of AML patients’ WB significantly increased the generation of mature/leukemia-derived DC, leading to increased frequencies of activated/memory T-cells and the provision of antigen-/leukemia-specific T or innate immune cell subtypes after T cell-enriched MLC, ultimately leading to improved blast cytotoxicity ex vivo in the context of relapses after allo-HCT.

## 4. Discussion

### 4.1. T-Cell Exhaustion Responsible for the Immunosuppressive Leukemic Microenvironment

Exhausted T-cells exhibit a reduced ability to proliferate in response to antigenic stimulation and show diminished production of key immunoregulatory cytokines, including interferon-gamma (IFNγ), tumor necrosis factor-alpha (TNF-α), and various interleukins. Additionally, the upregulation of ICM, such as PD1, CTLA4, LAG3, TIM3, KLRG1, TIGIT, and 2B4 on T-cells, contributes to T-cell exhaustion and impaired functionality, correlating with poor prognosis and reduced overall survival in AML patients [26,27,28,29]. In particular, high expression of PD1, CTLA4, TIM3, and TIGIT on CD8+ (memory) T cells is associated with relapse in AML patients following allo-HCT [12,27,29,30,31].

### 4.2. Higher Frequencies of ICM-Expressing Uncultured Blasts and T-Cells Found in AML Patients That Relapsed After Allo-HCT Compared to Healthy Donors

It is well known that the upregulation of PD1 on immune effector T-cells inhibits their effector function and promotes tumor progression, and that depletion of PD1-expressing T-cells enhances anti-tumor immunity [10,32]. However, the function of PD1 in suppressive immune cells and tumor cells is controversial [10].

Our data show and confirm [12,30,33] that higher frequencies of CTLA4- and PD1-expressing T-cells are found in uncultured AML patients’ samples compared to healthy blood donors’ samples, with high proportions of patients presenting with more than 20% of CTLA4 or PD1 expressing T-cells (Figure 1B). Notably, more T_CD3+CD4−_ cells expressing CTLA4, PD1, TIGIT, and TIM3 were found in these patients compared to healthy donors. This finding indicates that elevated frequencies of ICM-expressing cells may play a crucial role in leukemic relapse. We can add, in addition, that (in part significantly) higher frequencies of TIM3- and TIGIT-, but not LAG3-, KLRG1-, or 2B4-expressing T-cells were also found in patients vs. in healthy blood samples, thereby confirming previous results [4,7].

Patients presenting with high frequencies of CTLA4- or PD1-expressing T-cells were characterized by a worse response to induction therapy or survival [33]. Our data confirm these findings for CTLA4- but not for PD1-expressing T-cells in our patient cohort (relapse after allo-HCT) (Figure 4C). However, frequencies of CTLA4-, PD1-, 2B4-, TIM3-, or KLRG1-expressing T-cells, but not of LAG3- or TIGIT-expressing T-cells, correlated negatively with ex vivo achieved improved blast lysis (after T cell-enriched MLC; see Figure 4A and Appendix A) [6].

### 4.3. ICM-Expressing B- or NK Cells, DC, or Blasts

PD1 and CTLA4 are expressed not only on healthy T-, B-, or NK-cells but also on tumor-infiltrating immune cells, myeloid progenitor cells, and myeloid-derived suppressive cells (MDSCs), all of which contribute to the regulation of immune responses [10,11,12,13,32,34]. It was shown that PD1 can be expressed on NK or B cells, mediating impaired cytokine production, decreased cytolytic activity, and poor proliferation [35], and their function is restored after PD1 antibody treatment [36]. Moreover, not only (increased) PD1- and CTLA4-expressing T-cells, but also PD1- and CTLA4-expressing (circulating) disseminated cancer cells were detected in patients with solid tumors (e.g., melanoma; lung, breast, and pancreatic cancer); (Hodgkin/Non-Hodgkin) lymphoma or leukemia [10,12,33,37,38,39,40]. To date, the functions of PD1 and CTLA4 in AML have been predominantly investigated in T-cells, whereas their roles in AML blasts remain largely unexplored.

In our patient cohort (composed of patients who relapsed after SCT), we found low frequencies of TIM3, PDL2, and PDL1; however, we identified high average frequencies of PD1- and CTLA4-expressing blasts (Figure 1(A1,A2)). Whereas in most of the patients’ samples, less than 10% of the blasts co-expressed PD1 or CTLA4, in two patients, more than 60% of blasts were found to co-express CTLA4 or PD1 (Figure 1(A2) left side). In both patients, an average of 8.5% of T-cells co-expressed PDL1 and 9.6% co-expressed PDL2, indicating elevated expression of immune checkpoint ligands on T-cells.

The functional and prognostic relevance of PD1- or CTLA4-expressing tumor cells is controversial [12,41]: some groups have suggested that PD1-expressing tumor cells enhance tumor growth (by increasing the level of effector molecules responsible for cellular proliferation), whereas others showed a correlation between PD1-expressing lung cancer and inhibited proliferation (mediated by downregulated AKT and ERK1/2 phosphorylation). In these patients, a PD1 blockade enhanced tumor growth [41]. In a study by Bolkun et al. [33], PD1 or CTLA4 CD33+ (AML blast-containing) cells were detected. The frequencies of these cells were lower compared to our findings, which could be explained by the inclusion of all CD33+ (myeloid) cells in the “blast gate”, whereas we combined patients’ individual blast markers (e.g., CD117 or CD34) with PD1 or CTLA4. No significant differences in the frequencies of PD1- or CTLA4-expressing CD33+ (AML blast-containing) cells between patients with adverse vs. good ELN risk or patients who achieved or did not achieve complete remission after induction therapy were seen. Importantly, CTLA4-expressing blast cells appeared to serve as a potential marker for therapeutic response, with lower levels being observed in CR patients and elevated levels in NRs. This suggests a possible role for CTLA4 in mediating resistance to therapy.

Pistillo et al. [37] conducted flow cytometric analyses using a panel of anti-CTLA4 single-chain variable fragment (scFv) antibodies and demonstrated that CTLA4 was not detectable on the cell surface, but was abundantly expressed intracellularly in freshly isolated peripheral blood mononuclear cells (PBMCs), including T-, B-cells, CD34+ stem cells, and granulocytes. Surface expression of CTLA4 was inducible following stimulation with agents that specifically activate these cell types. It is plausible that CTLA4+ tumor cells may contribute to immune evasion by directly suppressing immune effector functions, thereby promoting disease progression and resistance to therapy [42,43,44].

Suo et al. [34] identified a subpopulation of AML cells characterized by high PD1 expression, which promoted leukemogenesis through activation of the MAPK/ERK signaling pathway. In another study, Strauss et al. demonstrated that myeloid-specific PD1 ablation was more effective in suppressing tumor growth compared to T cell-specific PD1 ablation. Cheng et al. [45] reported that aberrant PD1 activation in hematopoietic stem and progenitor cells (HSPCs) leads to increased hematopoietic cell death, bone marrow failure, and suppression of normal hematopoiesis, further highlighting the functional impact of PD1 signaling beyond the lymphoid compartment.

#### 4.3.1. Enhanced Intracellular Cytokine-Producing or Degranulating Immune Cells in Uncultured and Cultured (After MLC in Kit-M-Treated (vs. Untreated)) AML Patients and Healthy Donors’ WB

In uncultured cell samples, we observed that the addition of LAA (to patients’ samples) or SEB (to healthy donors’ samples) antigen stimulation resulted in increased frequencies of various degranulating and IFNγ-secreting cell subtypes (Appendix A). These findings further confirm that LAA stimulation can enhance the generation of leukemia-specific immunoreactive cells, as shown previously.

After MLC in Kit-M-treated AML patients’ samples, we confirmed preliminary data by showing increased IFNγ production by T cell subtypes (e.g., ß7-expressing T, T_non-naïve_, T_em/eff_ and T_cm_) in Kit-M-treated WB compared to controls (Appendix A) [17]. Additionally, SEB stimulation in healthy samples resulted in higher frequencies of IFNγ-producing immune cells. Additionally, we observed decreased frequencies of degranulating T_reg_ in Kit-M-treated WB vs. controls (Appendix A), thereby confirming previous results which suggest that Kit-M treatment enhances anti-leukemic responses in leukemic WB [19].

#### 4.3.2. DC-Based Immunotherapy

In this study, we confirm the successful conversion of leukemic blasts from AML/MDS patients who relapsed after allo-HCT into DC_leu_ as well as the transformation of healthy monocytes into DC. Both were established in Kit M-pretreated WB samples without inducing blast proliferation (Appendix A). Additionally, DCs express various ICMs, such as PD-L1 [46], CD86, and TIM3 [47], which modulate these interactions, influencing the balance between immune activation and tolerance. In this study, we also quantified ICM-expressing DCs, contributing to a better understanding of their role in regulating immune activity. Due to many cases with low blast counts (and, consequentially, non-detectable DC_leu_ counts in our cohort we could only quantify ICM/ICML-expressing DC. Previous studies confirmed that the potential antileukemic effects of Kit M treatment are independent of patients’ sex, age, ELN risk, HLA type, or stage of disease [19]. Additionally, Kit M pretreatment of samples from patients who relapsed after allo-HCT leads to reduced ICM/ICML (e.g., CTLA4, 2B4 et al.)-expressing DC compared to controls (Figure 2), which could be associated with reduced immunosuppressive effects in leukemia.

#### 4.3.3. Provision of Immunoreactive Cells and Increased Blastolytic Activity After MLC in Kit-M-Treated WB

As previously shown [19,25,48], we confirmed a generally higher activation status of immune cells after MLC in both healthy and patient samples pretreated with Kit M compared to those without pretreatment, leading to increased frequencies of activated cells and the generation of memory cells (Appendix A). These effects can, in part, be attributed to IL-2 stimulation, and this may indicate immunological activation against a variety of bacterial, viral, or mycotic targets [49,50,51]. After T-cell-enriched MLC, we also found, in Kit-M-pretreated patients’ WB, lower frequencies of ICM/ICML (e.g., TIGIT, CTLA4, PD1, 2B4, KLRG1) co-expressing T-cells compared to controls (Figure 3). We can state that the Kit-M-induced activation of immune cells after MLC occurs alongside a downregulation of ICM/ICML-expressing immune cells, which correlates with improved antileukemic activity (Figure 3 and Figure 4B).

We observed a higher number of cases with improved blast lysis achieved after T-cells-enriched MLC with Kit-M pretreated vs. not pretreated WB, following 3 or 24 h of incubation of blast targets with effector cells, relative to controls (Appendix A). These effects may be attributed to distinct, independent blastolytic mechanisms: the faster perforin/granzyme pathways, which lead to blast lysis predominantly after 3 h of co-incubation, and the slower Fas/FasL pathways, which result in blast lysis predominantly after 24 h of co-incubation [17,18,25,52].

### 4.4. Potential of ICM/ICML Monitoring

We identified negative correlations between improved blast lysis and frequencies of ICM/ICML (e.g., CTLA4, PD1, PDL2, 2B4, TIM3, KLRG1)-expressing uncultured T-cells (Figure 4A and Appendix A), suggesting that these immune cells have reduced blastolytic functionality. This implies that Kit M pretreatment may enhance antileukemic function and reverse ICM/ICML-mediated immunosuppression. However, after Kit M-pretreated MLC, we did not observe correlations between ICM/ICML-expressing T-cells with achieved blast lysis (Figure 4B and Appendix A). This suggests that Kit M enhances antileukemic activity ex vivo in an ICM-independent manner. Additional checkpoint-targeting antibodies in combination with Kit M (with/without treatment with hypomethylating agents (HMAs) with immune-modulating effects, as performed in these patients) could contribute to improved immune responses in the context of treatment of AML relapse post allo-HCT. Notably, ICM inhibition (e.g., CTLA4, PD1) alone or in combination with HMA (Decitabine, Azacitidine) for the treatment of AML relapse after allo-HCT showed only modest results [53,54,55]. However, subsequent translational analyses revealed that patients who presented with an exhausted T cell phenotype may benefit from these therapies.

In 14 patients, salvage therapy was based on HMA (Azacitidine or Decitabine) + Venetoclax. According to their individual responses to this line of therapy, patients were categorized as “Responders” or “Non-responders”. In unadjusted exploratory analyses, and with the caveat of small numbers, we found that lower frequencies of T-cells co-expressing CTLA4 and TIGIT correlated with a higher probability of patients responding to the salvage therapy (Figure 4C). The lack of statistical significance may be attributed to the small sample size. These findings suggest that higher frequencies of CTLA4- and TIGIT-expressing T-cells could be a predictor of response to HMA/Venetoclax-based salvage therapy. In addition, some studies reported [56,57] that HMA treatment induces the expression of ICM in AML cells. In contrast, our previous findings after individualized treatment of a relapsed AML patient after SCT treated with Kit M indicated in vivo generated mature DC_leu_, activation of leukemia-specific effector and memory cells, downregulation of several checkpoint marker-expressing T and NK cells, and of B_reg_ and Treg, along with a sustained clinical stabilization of the patient over 4 months [21]. Moreover, in vivo experiments involving leukemic rats, as well as an experimental clinical treatment of highly refractory AML patients before allo-HCT, showed leukemia-specific immune cell activation and stabilization of the disease [20].

These data illustrate the feasibility and tolerability of a “Kit M in vivo therapy” and its potential to modulate and improve antileukemic immunity in patients with (relapsed/refractory) AML before or after SCT.

In summary, an increase in ICM-expressing cells may play an important role in AML relapse following allo-HCT. In our study, we also found that Kit M treatment can reduce the frequencies of ICM-expressing T-cells, which correlate with leukemia-specific effects. This finding suggests that Kit M treatment could potentially eliminate ICM/ICML-expressing T-cells, thereby contributing to improved responses to salvage therapies following AML relapse post-allo-HCT.

## 5. Conclusions

Treatment of relapse of AML after allo-HCT remains a challenge. Only 20% of patients who respond to further chemotherapy achieve long-term remission with a second allo-HCT or with chemotherapy followed by donor lymphocyte infusions. We believe that analysis/quantification of ICM/ICML-expressing cells of several lines could become a very valuable parameter for assessing the efficacy of AML immunotherapy in future clinical applications. Kit M has potential for reinvigoration with immune cell-targeting approaches, improving the immune cells’ exhaustion status and thereby enabling the reactivation of antileukemic immune cells.

Understanding which mechanisms ultimately lead to AML relapse in each case would provide the basis for an effective therapy after disease recurrence. In fact, consensus on the epidemiology of the different relapse mechanisms and clear pathways for the translation into therapeutic decisions is lacking. But DC-based strategies can offer a potential method for AML patients after allo-HCT relapse.

## Figures and Tables

**Figure 1 cancers-17-02948-f001:**
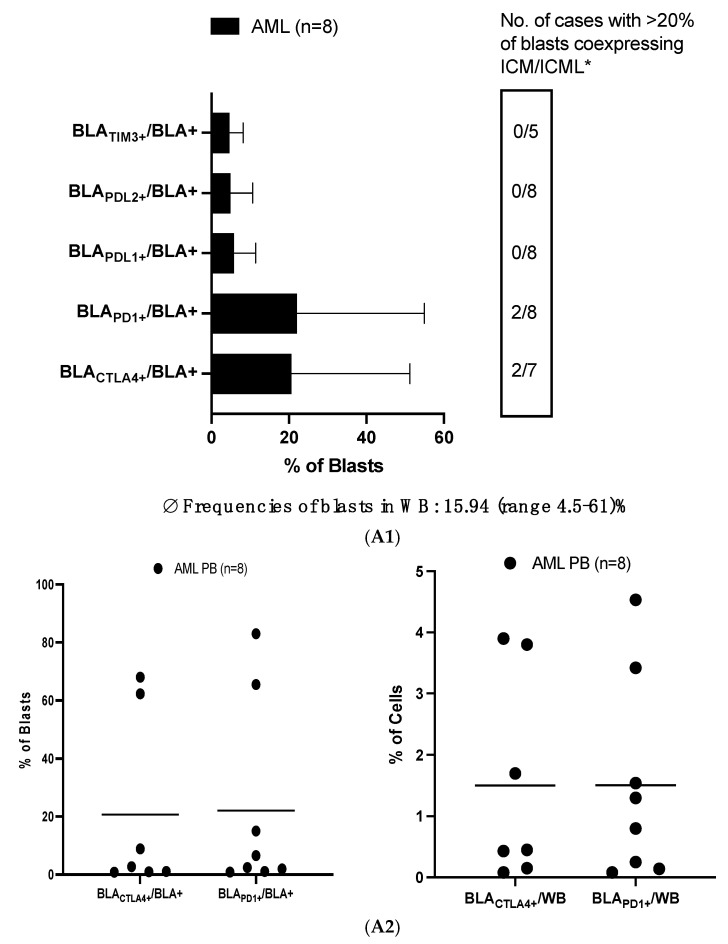
(**A1**) Frequencies of uncultured AML blasts co-expressing ICM/ICML. (**A2**) Frequencies of CTLA4 and PD1 expressing blasts in the blast fraction (left side) or in patients’ WB (right side). (**B**) High frequencies of uncultured T-cell (subtypes) from AML patients, but low frequencies of healthy T-cells co-express ICM (CTLA4 and PD1). Mean frequencies ± standard deviation of uncultured AML blasts (**A1**,**A2**) or T-cell subtypes (**B**) co-expressing ICM/ICML in AML patients’ and healthy donors’ WB, as evaluated by flow cytometry, are given. Data normality was assessed using the Shapiro–Wilk test. Statistical analyses were conducted using the Wilcoxon signed-rank test, with *p* ≤ 0.05 considered statistically significant. Double-sided arrows indicate (significant) differences between defined cell subtypes in AML patients and healthy donors. Abbreviations of cell types are given in Appendix A. * Number of cases with >20% blasts (**A1**) or T-cells (**B**) co-expressing ICM/ICML are given (box).

**Figure 2 cancers-17-02948-f002:**
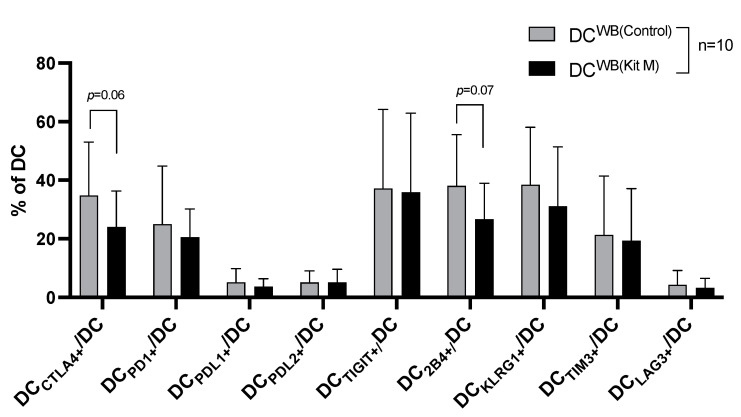
Comparable or lower frequencies of ICM/ICML expressing DC from AML WB with Kit-M-treated WB compared to Control. Mean frequencies ± standard deviation of ICM/ICML expressing DC subtypes from AML patients’ WB (with (DC^WB(Kit M)^) and without Kit M-pretreated WB (DC^WB(Control)^). Data normality was assessed using the Shapiro–Wilk test. Statistical analyses were conducted using *t*-tests: Differences were considered highly significant with *p* values ≤ 0.005, as significant with *p* values ≤ 0.05, and as borderline significant with *p* values between 0.05 and 0.10. Abbreviations of cell types are given in Appendix A.

**Figure 3 cancers-17-02948-f003:**
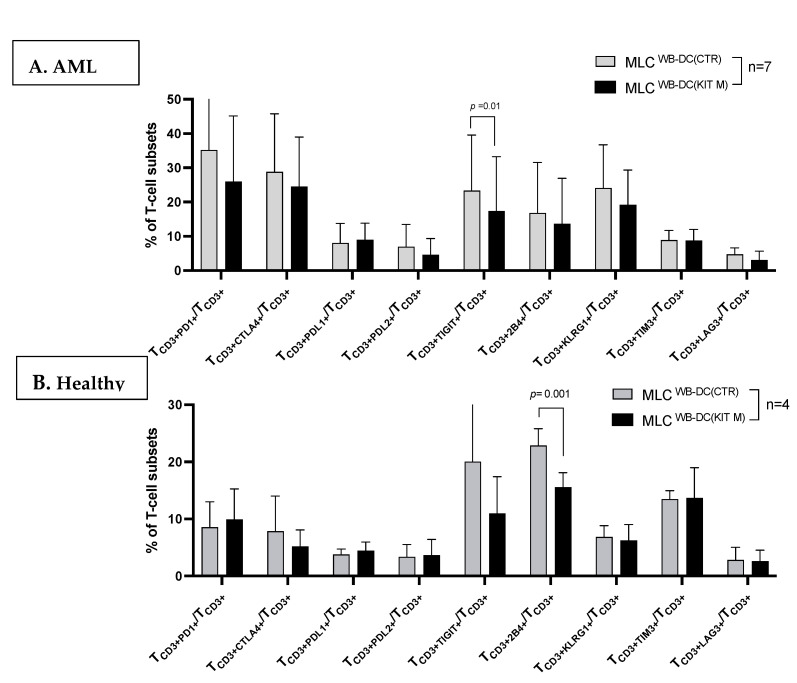
Downregulated ICM/ICML expressing AML and healthy T cells after MLC with Kit-M-treated WB compared to Control. Mean frequencies ± standard deviation of ICM/ICML expressing leukemic (**A**) or healthy (**B**) T cells compared to WB not pretreated with Kit M (MLC ^WB-DC(Control)^). Data normality was assessed using the Shapiro–Wilk test. Statistical analyses were conducted using *t*-tests: Differences were considered highly significant with *p* values ≤ 0.005, as significant with *p* values ≤ 0.05, and as borderline significant with *p* values between 0.05 and 0.10. Abbreviations of cell types are given in Appendix A.

**Figure 4 cancers-17-02948-f004:**
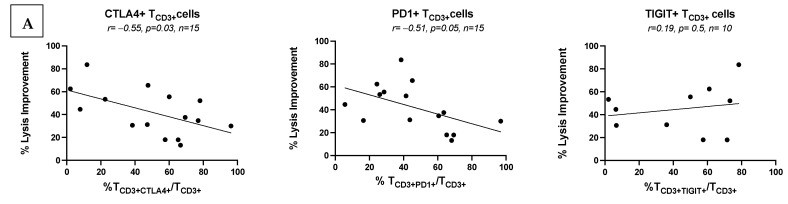
(**A**) Low frequencies of ICM/ICML expressing uncultured AML patients’ T-cells correlate with ex vivo later-achieved blast lysis and in vivo response to salvage therapy. (**B**) Frequencies of ICM/ICML-expressing cultured AML patients’ T-cells after Kit M-pretreated MLC did not correlate with ex vivo achieved blast lysis. (**C**) Low frequencies of ICM/ICML-expressing uncultured AML patients’ T-cells correlated with in vivo response to salvage therapy. R—correlation coefficient; *p*-significance; n—number of cases. Data normality was assessed using the Shapiro–Wilk test. Statistical analyses were conducted using *t*-tests: Differences were considered as significant with *p* values ≤ 0.05 and as borderline significant with *p* values between 0.05 and 0.10. Abbreviations of cell types are given in Appendix A.

## Data Availability

The data presented in this study are available in this article.

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
