# Peer review of "Dendritic Cells of Leukemic Origin (DCleu) Modulate the Expression of Inhibitory Checkpoint Molecules and Their Ligands on T Cells and Blasts in AML Relapse After Allogeneic Stem Cell Transplantation"

_cancers, 2025, doi:10.3390/cancers17182948_

Round 1
Reviewer 1 Report
Comments and Suggestions for Authors
AML is a hematological malignancy and the traditional clinical treatment, allo-HSCT could not complete cure patients. The relapses occur in 30-40% of AML patients with poor survivals. In the manuscript, the author focused the DC cells in AML patients. DC cells enable to decrease the immune checkpoints expression in AML patients’ T cells. It might improve the anti-leukemia effect for AML patients. However, the manuscript is not completed and persuasive. Most results are presented in the form of statistical results. The detection about T cell function from healthy donors or patients cocultured with DCs are lacked in vitro or in in vivo mouse model. It is indispensable to support the viewpoint of this manuscript. I don’t think the current version could be accepted. What's worse, some details needed to be checked carefully, and many mistakes still exist, especially about punctuation, such as, Line117, Line 171, Line 392, Line 440, et al.
The Figures are unshaped and incomplete. The Discussion Section is too long and unclear.
Author Response
Dear Reviewer of CANCERS,
Thank you for your reviews and your constructive comments. We have carefully revised the manuscript to address all the issues raised. Below, we provide a point-by-point response.
Comments of reviewers are given in bold letters. Our answers are given in standard letters.
Comments 1: Most results are presented in the form of statistical results. The detection about T cell function from healthy donors or patients cocultured with DCs are lacked in vitro or in in vivo mouse model. It is indispensable to support the viewpoint of this manuscript.
We thank the reviewer for the valuable comment highlighting the need to strengthen our conclusions regarding immune cells’ functionality after co-culture with leukemia-derived dendritic cells (DCleu). The application of functional cell assays (e.g., proliferation, cytokine production/release, degranulation or cytotoxicity assays) are important to confirm the immunostimulatory (DCleu mediated) antileukemic capacity after T cell enriched MLC from AML PB (or BM) sources -ex vivo or in vivo: We previously demonstrated that DCleu can be generated ex-vivo using the immune response modifiers GM-CSF and PGE-1 (“Kit-M”) and induce ( leukaemia-specific) B-, T and NK-cell responses in preclinical models (e.g: Klauer 2025; Unterfrauner et.al. 2023) or in vivo after treatment of leukemically diseased rats or therapy refractory patients before or after stem cell transplantation (Atzler, M et.al. IJMS 2024; Filippini VG et.al. Biomarker Research 2025). Thus, these studies support the core hypothesis of our manuscript—that DCleu , induced by Kit-M, possess functional antileukemic immune cell stimulatory capacity, both ex vivo and in vivo. Currently we start an in vivo mouse project: a syngenic leukemic cell line is transferred to mice and they are treated with (vs without) KIT-M.
We included some sentences in the discussion and include these references.
Supporting Literature:
Atzler, M., Baudrexler, T., Amberger, D. C., Rogers, N., Rabe, A., Schmohl, J., ... & Schmetzer, H. M. (2024). In vivo induction of leukemia-specific adaptive and innate immune cells by treatment of aml-diseased rats and therapy-refractory AML patients with blast modulating response modifiers. International Journal of Molecular Sciences, 25(24), 13469.
Unterfrauner M, Rejeski HA, Hartz A, Bohlscheid S, Baudrexler T, Feng X, et al. Granulocyte-Macrophage-Colony-Stimulating-Factor Combined with Prostaglandin E1 Create Dendritic Cells of Leukemic Origin from AML Patients' Whole Blood and Whole Bone Marrow That Mediate Antileukemic Processes after Mixed Lymphocyte Culture. Int J Mol Sci. 2023;24(24).
Filippini Velázquez, G., Anand, P., Abdulmajid, J., Feng, X., Weller, J. F., Hirschbuehl, K., Schmetzer, H., & Schmid, C. (in press). Clinical stabilization of a highly refractory acute myeloid leukaemia under individualized treatment with immune response modifying drugs by in vivo generation of dendritic cells of leukaemic origin (DCleu) and modulation of effector cells and immune escape mechanisms. *Biomarker Research*.
Comments 2: Some details needed to be checked carefully, and many mistakes still exist, especially about punctuation, such as, Line117, Line 171, Line 392, Line 440, et al.
We have thoroughly reviewed and corrected the manuscript for grammatical and punctuation errors, including the specific lines mentioned (Lines 117, 171, 392, 440, and others). We also performed a full language check to ensure clarity, consistency, and professional tone throughout the manuscript.
Comments 3: The Figures are unshaped and incomplete.
All figures have been restructured to ensure proper formatting and clarity. We have also double-checked all figure references in the text to ensure accuracy.
Comments 4: The Discussion Section is too long and unclear.
We have significantly revised the Discussion section. In this study, we analyzed the frequencies of various immune checkpoint molecule expressing cells in an ex vivo setting, comparing the AML cohort to healthy donors. Redundant and speculative content has been removed, and the key interpretations and conclusions have been clarified and condensed. Moreover, we have deleted the sub-chapter numbers.
We hope, that our intensively revised manuscript fulfills now criteria to be published in your journal. We are looking forward to a (positive) answer.
Best regards,
Xiaojia Feng
Reviewer 2 Report
Comments and Suggestions for Authors
The authors report a prospective study on ICM and Dendritic cells in relapsed AML. The topic is interesting and novel, although the sample size is very small and presentation of data and the reading of the paper is quite difficult.
I believe that the entire organization of presented data should be revised to become easier to read and to understand.
The introduction is complete and well written, the methods are detailed. The results are difficult to read, due to the structure of the paragraph. Too many subheadings and many figures make the content difficult to understand. Probably some of them might be moved to supplementary data. The discussion with many subheadings repeats the results, while the conclusions sound like a discussion.
Line 204: the authors state that samples with <5% blasts have been excluded, while the range of blasts is 4.5-61%: 4.5 is <5%
Line 511-512: "in part" has been repeated
Comments on the Quality of English LanguageThe English can be improved, some typing mistakes
Author Response
Dear Reviewer of CANCERS,
Thank you for your reviews and your constructive comments. We have carefully revised the manuscript to address all the issues raised. Below, we provide a point-by-point response.
Comments of reviewers are given in bold letters. Our answers are given in standard letters.
Comments 1: The results are difficult to read, due to the structure of the paragraph. Too many subheadings and many figures make the content difficult to understand. Probably some of them might be moved to supplementary data.
We have restructured the Results section to improve clarity and flow by reducing the number of subheadings and simplifying the paragraph structure. In addition, to simplify the main text, we have moved several supporting figures to the Supplementary Materials (now labeled as Supplementary Figure 2-7). This allows the core findings to stand out more clearly, while still making additional data available to readers.
Comments 2: The discussion with many subheadings repeats the results
We have substantially revised the Discussion section by removing repetitive descriptions of results and limiting the use of subheadings. The content is now more concise, with a clearer focus on interpretation and relevance.
Comments 3: The conclusions sound like a discussion.
We have significantly revised the Conclusion section to avoid overlap with the Discussion.
Comments 4: Line 204: the authors state that samples with <5% blasts have been excluded, while the range of blasts is 4.5-61%: 4.5 is <5%.
We thank the reviewer for catching this inconsistency. The text has been corrected to accurately reflect the inclusion criteria. Specifically, we have amended the statement to indicate that blast data of the one patient presenting with ≤4% blasts were excluded. The remaining samples presented with more than 4.5% blasts and could be evaluated.
Comments 5: Line 511-512: "in part" has been repeated
We have corrected the sentence and removed the redundant phrase.
We hope, that our intensively revised manuscript fulfills now criteria to be published in your journal. We are looking forward to a (positive) answer.
Best regards,
Xiaojia Feng
Reviewer 3 Report
Comments and Suggestions for Authors
In the manuscript entitled " Dendritic cells from leukemic origin (DCleu) modulate the expressions of inhibitory checkpoint molecules and their ligands on T cells and Blasts in AML relapse after allogeneic stem cel transplantation", the authors induce dendritic cells of leukemic origin („ex vivo”) and evaluate their influence on the expression of negative immune checpoint molecules and their ligands on blasts and T cells in AML relapse post-allo-HSCT.
Based on the manuscript, I have the following comments and questions:
- The Introduction section lacks information about MDS. It is also recommended that the INCYT and DEG test descriptions be included in the 'Materials and Methods' section.
- One of the aims of this study was to demonstrate the generation of leukemic- DC subpopulations; however, these results are missing.
- What were the inclusion and exclusion criteria for patients in both the study and control groups?
- The methods are described in a laconic and imprecise manner. For instance, there is no information regarding the immunophenotyping procedure and the monoclonal antibodies employed for evaluating immune cells.
- Tables are missing from the article and supplement.
- Why were leukemic-DCs generated instead of sorting them from the patients and stimulating them e.x. LAA? Furthermore, how was checked the purity of DCs obtained using Kit M verified?
- The statistical analyses primarily used t-tests. Please provide the results of the normality tests that informed the decision to use a parametric test.
- In the Results section of Figure 1B it is suggested to also use CD - cluster of diffentation, TCD3+ PDL2+/TCD3+
- The descriptions of results 3.3, 3.4.1., 3.5, 3.6, 3.7, 3.8 are too general and do not contain statistical data.
- What was the ratio of leukemic-DCs to MLR lymphocyte cultures, and how long after administration was ICM/ICML expression assessed? What were the culture conditions?
- The discussion is too long and general, and repeats content covered in the introduction. Please focus on providing a thorough analysis and interpretation of the results obtained.
Author Response
Dear Reviewer of CANCERS,
We are very grateful to your careful evaluation and thoughtful suggestions, which have greatly improved the quality and clarity of our manuscript. Below, we respond to each point in detail and summarize the corresponding changes made in the revised version.
Comments of reviewers are given in bold letters. Our answers are given in standard letters.
- The Introduction section lacks information about MDS.
We included only one patient diagnosed with MDS in our study. Given the overlap between MDS and AML in terms of pathogenesis and clinical relevance—we considered this case within the broader AML context. To maintain focus and clarity in our introduction, we chose not to elaborate on MDS specifically. However, we are happy to add a brief clarification if the reviewer deems it necessary.
- It is also recommended that the INCYT and DEG test descriptions be included in the 'Materials and Methods' section.
We agree and have now added a detailed description of the INCYT and DEG test in the Supplementary Materials.
- One of the aims of this study was to demonstrate the generation of leukemic- DC subpopulations; however, these results are missing.
Thank you for pointing this out. These results are now included in the revised Results section 3.3.1 and presented in Suppl.Figure 3.
- What were the inclusion and exclusion criteria for patients in both the study and control groups?
The inclusion criteria included patients with relapsed AML/MDS after allo-HCT, with healthy donors as the control group. This has been incorporated in the Materials and Methods section under “2.2 Patients”.
- The methods are described in a laconic and imprecise manner. For instance, there is no information regarding the immunophenotyping procedure and the monoclonal antibodies employed for evaluating immune cells.
We have added a detailed description of the immunophenotyping protocol, including the staining procedures, the panel of monoclonal antibodies used, the flow cytometer model, and the data analysis software. These details appear in the Supplementary Materials.
- Tables are missing from the article and supplement.
We have now included 2 tables summarizing patient demographics (Table 1), antibody panels (Table 2). All tables are properly referenced in the text and included in Supplementary Materials.
- Why were leukemic-DCs generated instead of sorting them from the patients and stimulating them e.x. LAA? Furthermore, how was checked the purity of DCs obtained using Kit M verified?
We thank the reviewer for this important methodological question. Kit M (containing clinically approved drugs GM-CSF and PGE1) directly targets blasts and converts them to leukemia derived DC (DCleu), eliminating the need for LAA loading on (ex vivo generated monocyte derived) DC followed by purification steps and adoptive retransfer to patients. Our approach is designed with clinical application of ‘Kit M derived drugs’ in mind, as it allows for in vivo conversion of leukemic blasts into DCleu by the patients themselves, thereby bypassing complex ex vivo procedures such as cell sorting, antigen loading, and adoptive cell transfer. Regarding purity, the DCleu population obtained using Kit M was quantified using standard flow cytometry markers for DC and blasts, ensuring accurate identification and characterization.
- The statistical analyses primarily used t-tests. Please provide the results of the normality tests that informed the decision to use a parametric test.
We have added the results of Shapiro–Wilk normality tests to justify the use of parametric tests. For data that did not meet the assumptions of normality (especially with small sample sizes), we reanalyzed the data using non-parametric alternatives (e.g., Wilcoxon signed-rank). This is described in the revised Statistical Analysis section (page X, Figure1B), and results have been updated in the relevant figure legends.
- In the Results section of Figure 1B it is suggested to also use CD - cluster of diffentation, TCD3+ PDL2+/TCD3+
We have clarified the naming conventions in all Figures and throughout the manuscript to include full definitions for consistency and clarity. This adjustment is also reflected in the revised figure legend.
- The descriptions of results 3.3, 3.4.1., 3.5, 3.6, 3.7, 3.8 are too general and do not contain statistical data.
The results presented in Sections 3.2, 3.3.1, 3.4.1, 3.5 and 3.6 were addressed intensively and published in earlier studies from our lab. In the context of the manuscript presented here we just used the experiments to confirm, that the strategy worked in the settings presented here (using relapsed samples after SCT). Therefore, the data are given in the supplement and new data generated are given in detail in Sections 3.1, 3.3.2, 3.4.2 and 3.7, where we offer a comprehensive analysis and interpretation.
- What was the ratio of leukemic-DCs to MLC lymphocyte cultures, and how long after administration was ICM/ICML expression assessed? What were the culture conditions?
The leukemic-DC to lymphocyte ratio in MLC experiments was 1:10. Frequencies of ICM/ICML expressing cells was assessed at 3 time points: Day 0 (Baseline, on uncultured cells- Blasts/T cells); Day 7 (after DC culture, on DC), and Day 14 (after MLC, on T cells).
DC/MLC were cultured under physiological conditions (37 â—¦C, 5% CO2, 21% O2 and 95% humidity). These methodological details have now been added to the revised Supplementary Materials.
- The discussion is too long and general, and repeats content covered in the introduction. Please focus on providing a thorough analysis and interpretation of the results obtained.
The Discussion has been substantially revised and shortened. Sub chapter numbers are deleted. Repetitive content was removed, and the section now focuses more specifically on interpreting our findings.
We hope, that our intensively revised manuscript fulfills now criteria to be published in your journal. We are looking forward to a (positive) answer.
Best regards,
Xiaojia Feng
Round 2
Reviewer 2 Report
Comments and Suggestions for Authors
The authors have revised the manuscript according to general suggestions.
The manuscript is more flowing and the comprehension is easier.
The small sample size remains a limitation of the paper, however I believe it is acceptable in this version. Whenever possible please make the discussion mor organica with less subheadings.
Author Response
Comments: Please make the discussion more organic with less subheadings.
Dear Reviewer:
Thank you for your suggestion. In the discussion section, we first combine the two parts: "Enhanced intracellular cytokine production or degranulation of immune cells in uncultured patients and healthy donors´WB" and "Enhanced intracellular cytokine production or degranulation of immune cells after MLC in Kit-M-treated (vs. untreated) AML and healthy Donors´WB " into a single section.
Second, we combined : "Provision of immunoreactive cells after MLC in Kit-M-treated WB" and "increased blastolytic activity after MLC in Kit-M-treated WB" into another single section.
Finally, we deleted the last subtitle "Innate and adaptive immune system and antileukemic activity"
The revised structure is now more logical and simplified.
We hope that the revisions have adequately addressed your comment and we are looking forward to your answer.
Best regards,
Xiaojia Feng
Reviewer 3 Report
Comments and Suggestions for Authors
The authors have sufficiently addressed most of my comments, but I also suggest presenting an example of flow cytometry gating on T, B, or NK cells, dendritic cells, monocytes or blasts expressing ICM/ICML.
Author Response
Comments: Presenting an example of flow cytometry gating on T, B, or NK cells, dendritic cells, monocytes or blasts expressing ICM/ICML.
Dear Reviewer,
Thank you for your suggestion.
In our supplementary materials, we have shown an example of CTLA4 and PD1 expressing on Blasts (Suppl. Figure 1A). In addition, we have now included an additional gating example of ICM on T cells and T, B, NK, DC, monocytes-gating-Graph (Suppl. Figure 1B). In our study, we did not measure ICM/ICML expressing B, NK or monocytes.
We hope that the revisions have adequately addressed your comment and we are looking forward for your answer.
Best regards,
Xiaojia Feng